# Revisiting Unbiased Implicit Variational Inference

**Tobias Pielok** [1 2]   **Bernd Bischl** [1 2]   **David Rügamer** [1 2]

## Abstract

Recent years have witnessed growing interest in semi-implicit variational inference (SIVI) methods due to their ability to rapidly generate samples from complex distributions. However, since the likelihood of these samples is non-trivial to estimate in high dimensions, current research focuses on finding effective SIVI training routines. Although unbiased implicit variational inference (UIVI) has largely been dismissed as imprecise and computationally prohibitive because of its inner MCMC loop, we revisit this method and show that UIVI's MCMC loop can be effectively replaced via importance sampling and the optimal proposal distribution can be learned stably by minimizing an expected forward Kullback–Leibler divergence without bias. Our refined approach demonstrates superior performance or parity with state-of-the-art methods on established SIVI benchmarks.

## 1. Introduction

Bayesian inference, such as sampling-based or variational inference, is an important foundation for constructing uncertainty quantification measures for machine learning models. In variational inference (VI), samples are generated from a target distribution function $p_{\boldsymbol{z}}$ with the associated random variable $\boldsymbol{z}$, which can only be evaluated but not directly sampled from and is possibly unnormalized. This could be, e.g., a Bayesian posterior distribution or the canonical distribution w.r.t. a physical system. For this, a family $\mathcal{Q}_{\boldsymbol{z}}$ over distributions with a tractable sampling procedure is chosen, and a divergence measure $D$ where $D$ quantifies the dissimilarity between two distributions. The target distribution $p_{\boldsymbol{z}}$ can then be approximated by finding $q_{\boldsymbol{z}}^* \in \mathcal{Q}_{\boldsymbol{z}}$ which is closest to $p_{\boldsymbol{z}}$ w.r.t. $D$, i.e., $q_{\boldsymbol{z}}^* \in \arg\min_{q_{\boldsymbol{z}} \in \mathcal{Q}_{\boldsymbol{z}}} D(q_{\boldsymbol{z}}, p_{\boldsymbol{z}})$.

---

[1]Department of Statistics, LMU Munich, Munich, Germany [2]Munich Center for Machine Learning (MCML), Munich, Germany. Correspondence to: Tobias Pielok <tobias.pielok@stat.uni-muenchen.de>.

*Proceedings of the $42^{nd}$ International Conference on Machine Learning*, Vancouver, Canada. PMLR 267, 2025. Copyright 2025 by the author(s).

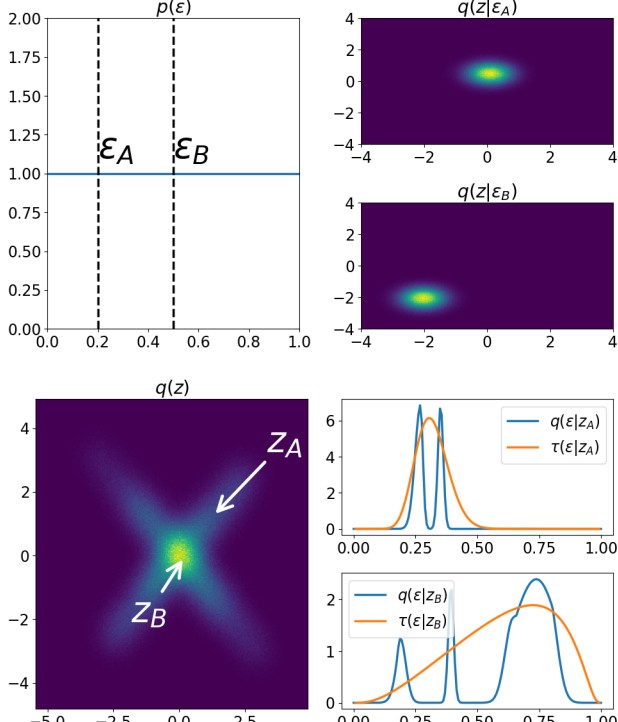

*Figure 1.* We sample from a semi-implicit distribution $q(\boldsymbol{z})$ by sampling from the latent distribution $p(\boldsymbol{\epsilon})$ and subsequently from the conditional distribution $q(\boldsymbol{z}|\boldsymbol{\epsilon})$. The simple distributions $p(\boldsymbol{\epsilon})$ and $q(\boldsymbol{z}|\boldsymbol{\epsilon})$ can induce a complicated distribution $q(\boldsymbol{z})$ and consequently a potentially even more complicated reverse conditional distribution $q(\boldsymbol{\epsilon}|\boldsymbol{z})$. AISIVI learns a mass-covering representation $\tau(\boldsymbol{\epsilon}|\boldsymbol{z})$ of $q(\boldsymbol{\epsilon}|\boldsymbol{z})$ to estimate $\nabla_{\boldsymbol{z}} \log q(\boldsymbol{z})$ in high dimensions.

### 1.1. Implicit Variational Inference

In contrast to VI, where we assume that $q_{\boldsymbol{z}} \in \mathcal{Q}_{\boldsymbol{z}}$ is an explicit distribution, i.e., we can evaluate $q_{\boldsymbol{z}}$, for implicit VI (IVI) we have no direct access to $q_{\boldsymbol{z}}$ and can only produce samples from $q_{\boldsymbol{z}}$, i.e., $q_{\boldsymbol{z}}$ is an implicit distribution. Representative examples of explicit and implicit distributions are normalizing flows (NFs) and neural samplers, which transform a random variable via an arbitrary neural network (NN), respectively. While NFs can be trained stably, they are known to smooth out sharp target distributions. In contrast, neural samplers can model highly complex and sharp distributions but are notoriously hard to train. This naturally suggests combining them.

Semi-implicit variational inference (SIVI; Yin & Zhou, 2018) offers a compromise between VI and IVI. Since we sample from semi-implicit distribution $q_z$ by sampling the parameters $y$ of an explicit distribution[1] $q_{z|y}$ from an implicit distribution $q_y$, its representative capabilities come close to those of an implicit distribution, but $q_z$ of a semi-implicit distribution can be estimated in a principle manner.

More formally, assuming that the target $z \sim p_z$ is a continuous random variable taking values in $Z \subseteq \mathbb{R}^{d_Z}$ where $d_Z \in \mathbb{N}$, we approximate its probability density function via an uncountable mixture of densities s.t.

$$q_z(z) = \mathbb{E}_{y \sim q_y} \left[ q_{z|y}(z|y) \right]. \tag{1}$$

For SIVI, the random variable $y$ taking values in $Y \subseteq \mathbb{R}^{d_Y}$ where $d_Y \in \mathbb{N}$ is drawn via a neural sampler, i.e.,

$$\epsilon \sim p_\epsilon \Rightarrow y = f_\phi(\epsilon) \tag{2}$$

where $\epsilon$ is a latent random variable taking values in $E \subseteq \mathbb{R}^{d_E}$ where $d_E \in \mathbb{N}$ and $f_\phi : E \to Y$ is a NN with parameters $\phi \in \mathbb{R}^{d_\phi}$ where $d_\phi \in \mathbb{N}$. Consequently, since every $\phi$ defines $q_z$, the distribution family $\mathcal{Q}_z$ is also parametrized by the NN parameters $\phi$. With Eq. 1 and Eq. 2, we also directly get that

$$q_z(z) = \mathbb{E}_{\epsilon \sim p_\epsilon} \left[ q_{z|\epsilon}(z|\epsilon) \right] \tag{3}$$

where $q_{z|\epsilon}(z|\epsilon) = q_{z|y}(z|f_\phi(\epsilon))$.

### 1.2. Our Contributions

In this work, we focus on the efficient estimation of the score gradient $\nabla_z \log q_z$, which enables us to train SIVI models even in high dimensions. For this, we propose using importance sampling (IS) with an adaptively informed proposal distribution $\tau_{\epsilon|z}$ modeled by a conditional normalizing flow (CNF). We show that $\tau_{\epsilon|z} = q_{\epsilon|z}$ debiases our score gradient estimate and propose a stable training routine of the CNF via an expected forward Kullback-Leibler divergence. Our contribution advances both mathematical insights of SIVI and contributes two new algorithms.

## 2. Background and Missed Opportunities

### 2.1. Reparametrizable Semi-implicit Distributions

In this work, we assume[2] that the reparametrization trick (Kingma & Welling, 2014) is applicable to $q_{z|y}$, i.e., there

---

[1]usually a common, unimodal distribution such as, e.g., a normal distribution

[2]We could even lessen our assumption by only assuming that implicit reparametrization gradients can be computed (Figurnov et al., 2018), but this is not the focus of this paper.

exist a random variable $\eta$ taking values in $H \subset \mathbb{R}^{d_H}$ where $d_H \in \mathbb{N}$ and a differentiable function $g : Y \times H \to Z$ s.t.

$$\epsilon, \eta \sim p_{\epsilon,\eta} \Rightarrow \underbrace{g(f_\phi(\epsilon), \eta)}_{=:h_\phi(\epsilon,\eta)} \sim q_z \tag{4}$$

where $p_{\epsilon,\eta}$ is the joint distribution of the independent random variables $\epsilon$ and $\eta$ which does not depend on $\phi$. From this, it directly follows that

$$\mathbb{E}_{z \sim q_z} [a_\phi(z)] = \mathbb{E}_{\epsilon,\eta \sim p_{\epsilon,\eta}} [a_\phi(h_\phi(\epsilon, \eta))] \tag{5}$$

where $a_\phi(z) : Z \to \mathbb{R}$ is a differentiable function with parameters $\phi$. Hence, under our assumptions, the reparametrization trick can be applied to $q_z$.

### 2.2. Path gradient estimator and $D_{\mathrm{KL}}$ minimization

We choose to minimize the reverse Kullback-Leibler divergence $D_{\mathrm{KL}}$, i.e.,

$$D_{\mathrm{KL}}(q_z \| p_z) = \mathbb{E}_{z \sim q_z} \left[ \log \left( \frac{q_z(z)}{p_z(z)} \right) \right]. \tag{6}$$

On the one hand, one of the main advantages of $D_{\mathrm{KL}}$ is that if we can evaluate $q_z$ we can compute unbiased estimates of the gradients w.r.t. the parameters $\phi$ of $q_z$, which is especially useful when stochastic gradient descent methods are employed to minimize the objective. On the other hand, the reverse $D_{\mathrm{KL}}$ is known to underestimate the variance if the variational distribution $q_z$ is not sufficiently expressive (Andrade, 2024). However, for SIVI, this is rarely relevant, as the variational distribution $q_z$ is highly expressive due to its implicit nature.

Since $q_z$ is amenable to the reparametrization trick, we can follow Roeder et al. (2017) to formulate a low-variance gradient estimator of $D_{\mathrm{KL}}$, the so-called path gradient estimator

$$\nabla_\phi D_{\mathrm{KL}}(q_z \| p_z) =$$
$$\mathbb{E}_{\epsilon,\eta \sim p_{\epsilon,\eta}} \left[ \nabla_z \left( \log q_z(z) - \log p_z(z) \right) \Big|_{z=h_\phi(\epsilon,\eta)} \tag{7} \right.$$
$$\left. \cdot \nabla_\phi h_\phi(\epsilon, \eta) \right].$$

While this result also appeared in the context of SIVI as an intermediate result in Titsias & Ruiz (2019), its far-reaching implications were not discussed since this expression was not of interest for the authors' final derivation (see Section 2.3). Not only does the path gradient estimator in Eq. 7 reduce the variance of the gradient estimation, but it also vastly reduces the computational demand in contrast to the reparametrization trick, for which we would need to estimate the gradient

$$\nabla_\phi \log q_z(h_\phi(\epsilon, \eta)) =$$
$$\nabla_\phi \log \left[ \mathbb{E}_{\tilde{\epsilon} \sim p_\epsilon} \left[ q_{z|y}(h_\phi(\epsilon, \eta)|f_\phi(\tilde{\epsilon})) \right] \right]. \tag{8}$$

This simple observation leads to a surprisingly well-performing approach, which we will introduce in Section 3.

## 2.3. Unbiased Implicit Variational Inference

The problematic term of the path gradient estimator in Eq. 7 is the score gradient $\nabla_{\boldsymbol{z}} \log q_{\boldsymbol{z}}(\boldsymbol{z})$, for which no analytical expression exists. Titsias & Ruiz (2019) proved for UIVI that

$$\mathbb{E}_{\boldsymbol{\epsilon} \sim q_{\boldsymbol{\epsilon}|\boldsymbol{z}}} \left[ \nabla_{\boldsymbol{z}} \log q_{\boldsymbol{z}|\boldsymbol{\epsilon}}(\boldsymbol{z}|\boldsymbol{\epsilon}) \right] = \nabla_{\boldsymbol{z}} \log q_{\boldsymbol{z}}(\boldsymbol{z}), \quad (9)$$

i.e., if we can produce samples from the intractable conditional distribution $q_{\boldsymbol{\epsilon}|\boldsymbol{z}}$, we can compute an unbiased estimate of the score gradient $\nabla_{\boldsymbol{z}} \log q_{\boldsymbol{z}}(\boldsymbol{z})$. Titsias & Ruiz (2019) propose to sample $\boldsymbol{z}, \boldsymbol{\epsilon} \sim q_{\boldsymbol{z}, \boldsymbol{\epsilon}}$ and use MCMC with target distribution[3] $q_{\boldsymbol{\epsilon}|\boldsymbol{z}}$. The MCMC chains are initialized at $\boldsymbol{\epsilon}$ because it already stems from the stationary distribution $q_{\boldsymbol{\epsilon}|\boldsymbol{z}}$. However, we can not use $\boldsymbol{\epsilon}$ directly since this would violate the independence assumption, which is needed for an unbiased estimate in Eq. 9. Therefore, MCMC has to run as long as the sample produced by the $i$-th chain $\boldsymbol{\epsilon}'_i$ is independent of $\boldsymbol{\epsilon}$. Titsias & Ruiz (2019) argue that only a few steps of MCMC are needed since the chains are already initialized at the stationary distribution. However, as it can be seen in Figure 1, $q_{\boldsymbol{\epsilon}|\boldsymbol{z}}$ is likely multimodal with regions of vanishing probability potentially occurring between the modes due to the implicit and possibly very complicated nature of $q_{\boldsymbol{z}}$. In such cases, very long chains would be needed to effectively break the dependence between $\boldsymbol{\epsilon}$ and $\boldsymbol{\epsilon}'_i$, rendering the already computationally intensive method as prohibitive. Furthermore, note that the number of chains cannot reduce the bias introduced by the prevailing dependence between $\boldsymbol{\epsilon}$ and $\boldsymbol{\epsilon}'_i$.

In light of these observations, we propose a novel method in Section 3 to fix the encountered shortcomings.

## 2.4. Conditional Normalizing Flows

Normalizing flows (NF; see, e.g., Papamakarios et al., 2021) leverage the change of variable method to model complex distributions by repeatedly transforming a random variable stemming from a simple error distribution. More specifically, for a random variable $\boldsymbol{u}$ taking values in $U \subseteq \mathbb{R}^{d_U}$ where $d_U \in \mathbb{N}$ and a differentiable and invertible transformation $T_{\boldsymbol{\theta}} : U \to U$ with parameters $\boldsymbol{\theta} \in \Theta \in \mathcal{R}$ it holds that

$$\boldsymbol{u} \sim p_{\boldsymbol{u}}, \boldsymbol{\epsilon} = T_{\boldsymbol{\theta}}(\boldsymbol{u}) \Rightarrow \quad \boldsymbol{\epsilon} \sim q_{\boldsymbol{\epsilon}},$$
$$q_{\boldsymbol{\epsilon}}(\boldsymbol{\epsilon}) = p_{\boldsymbol{u}}(T_{\boldsymbol{\theta}}^{-1}(\boldsymbol{\epsilon})) \left| \det J_{T_{\boldsymbol{\theta}}^{-1}}(\boldsymbol{\epsilon}) \right| \quad (10)$$

---

where $J_{T_{\boldsymbol{\theta}}^{-1}}$ is the Jacobian of the inverse function of $T_{\boldsymbol{\theta}}$. A *conditional* NF (CNF) is a differentiable map $T_{\boldsymbol{\theta}} : U \times Z \to U, (\boldsymbol{\epsilon}, \boldsymbol{z}) \mapsto T_{\boldsymbol{\theta}}(\boldsymbol{\epsilon}, \boldsymbol{z})$ such that for every $\boldsymbol{z} \in Z$ it holds that $T_{\boldsymbol{\theta}}(\cdot, \boldsymbol{z})$ is a NF.

A plethora of different NFs have been proposed over the last years. In this work, we use affine coupling layers as introduced in RealNVP (Dinh et al., 2017) because sampling and evaluating their likelihood is equally computationally efficient, and they can be scaled up to high dimensions (Andrade, 2024). Specifically, we use a conditional variant of affine coupling layers similar to one introduced in Lu & Huang (2020). While this is a natural choice, other combinations could provide additional performance gains as also discussed in Section 6.

## 3. Method

Starting from Eq. 7 we can rewrite the problematic score term, i.e.,

$$\nabla_{\boldsymbol{z}} \log q_{\boldsymbol{z}}(\boldsymbol{z}) = \nabla_{\boldsymbol{z}} \log \left[ \mathbb{E}_{\boldsymbol{\epsilon} \sim p_{\boldsymbol{\epsilon}}} \left[ q_{\boldsymbol{z}|\boldsymbol{\epsilon}}(\boldsymbol{\epsilon}) \right] \right]. \quad (11)$$

Thus, a straightforward Monte Carlo (MC) estimator of the score gradient is

$$s_{\mathrm{MC},k}(\boldsymbol{z}) = \nabla_{\boldsymbol{z}} \log \left( \frac{1}{k} \sum_{i=1}^{k} q_{\boldsymbol{z}|\boldsymbol{\epsilon}}(\boldsymbol{z}|\boldsymbol{\epsilon}_i) \right), \quad (12)$$

where $(\boldsymbol{z}, \boldsymbol{\epsilon}_1) \sim q_{\boldsymbol{z}, \boldsymbol{\epsilon}}$ and $\boldsymbol{\epsilon}_i \stackrel{\text{i.i.d.}}{\sim} p_{\boldsymbol{\epsilon}}, i = 2, \ldots, k$. This is a consistent estimator of the score gradient $\nabla_{\boldsymbol{z}} \log q_{\boldsymbol{z}}(\boldsymbol{z})$ and for large $k$ its bias

$$\mathbb{E}_{\boldsymbol{\epsilon}_i \sim p_{\boldsymbol{\epsilon}}} \left[ \nabla_{\boldsymbol{z}} s_{\mathrm{MC},k}(\boldsymbol{z}) \right] - \nabla_{\boldsymbol{z}} \log \left( q_{\boldsymbol{z}}(\boldsymbol{z}) \right) \approx$$
$$- \nabla_{\boldsymbol{z}} \left( \frac{\mathbb{V}_{\boldsymbol{\epsilon} \sim p_{\boldsymbol{\epsilon}}} \left[ q_{\boldsymbol{z}|\boldsymbol{\epsilon}}(\boldsymbol{z}|\boldsymbol{\epsilon}) \right]}{2(k-1) \cdot q_{\boldsymbol{z}}(\boldsymbol{z})^2} \right) \quad (13)$$

(see Appendix A.1 for the proof). Note that including $\boldsymbol{\epsilon}_1$ introduces additional bias but strongly reduces the variance since $\boldsymbol{\epsilon}_1 \sim q_{\boldsymbol{\epsilon}|\boldsymbol{z}}$. A closely related estimator was derived in Molchanov et al. (2019), but their estimator is purely based on the reparametrization trick and does not benefit from the advantages discussed in Section 2.2 and Section 3.2.

Although we would expect that for high dimensions, the contribution of $q_{\boldsymbol{z}|\boldsymbol{\epsilon}}(\boldsymbol{z}|\boldsymbol{\epsilon}_i)$ resulting from uninformed $\boldsymbol{\epsilon}_i$ to be nearly negligible to our estimator, $s_{\mathrm{MC},k}$ performs surprisingly well.

Based on the previous observation, we devise a new importance sampling (IS) version of Eq. 11, given as follows:

$$\nabla_{\boldsymbol{z}} \log q_{\boldsymbol{z}}(\boldsymbol{z}) =$$
$$\nabla_{\boldsymbol{z}} \log \left( \mathop{\mathbb{E}}_{\boldsymbol{\epsilon} \sim \tau_{\boldsymbol{\epsilon}|\tilde{\boldsymbol{z}}}} \left[ \frac{p_{\boldsymbol{\epsilon}}(\boldsymbol{\epsilon}) q_{\boldsymbol{z}|\boldsymbol{\epsilon}}(\boldsymbol{z}|\boldsymbol{\epsilon})}{\tau_{\boldsymbol{\epsilon}|\tilde{\boldsymbol{z}}}(\boldsymbol{\epsilon}|\tilde{\boldsymbol{z}})} \right] \right) \Bigg|_{\tilde{\boldsymbol{z}}=\boldsymbol{z}}. \quad (14)$$

The idea of enhancing SIVI with importance sampling was also proposed by Sobolev & Vetrov (2019), but their approach is more expensive than ours due to the joint optimization of the proposal distribution and the SIVI model, rendering more expressive conditional models, such as CNFs, infeasible in practice.

**Importance Sampling Estimator** Based on Eq. 14, we can estimate $\nabla_z \log q_z(z)$ using the following score gradient estimator

$$s_{\mathrm{IS},k}(z) = \nabla_z \log \left( \frac{1}{k} \sum_{i=1}^{k} \frac{p_\epsilon(\epsilon_i) q_{z|\epsilon}(z|\epsilon_i)}{\tau_{\epsilon|\widetilde{z}}(\epsilon_i|\widetilde{z})} \right) \Bigg|_{\widetilde{z}=z}, \quad (15)$$

where $\epsilon_i \sim \tau_{\epsilon|z}, i = 1, \ldots, k$. We show in Appendix A.2 that this estimator is consistent when $\mathrm{supp}(q_{\epsilon|z}) \subset \mathrm{supp}(\tau_{\epsilon|z})$. To also make this estimator efficient, we need to generate samples $\tau_{\epsilon|z}$ and evaluate their likelihood efficiently. A suitable option in this case is to model $\tau_{\epsilon|z}$ with a sequence of conditional affine coupling layers (see Section 2.4).

Since we optimize $\tau_{\epsilon|z}$ and $q_z$ alternately, we are interested in the optimal $\tau_{\epsilon|z}$ for a fixed $q_z$. This leads us to the following proposition:

**Proposition 3.1.** *Choosing $\tau_{\epsilon|z} = q_{\epsilon|z}$ debiases our proposed score gradient estimate $s_{\mathrm{IS},k}$, i.e.,*

$$\mathbb{E}_{\epsilon_i \sim q_{\epsilon|z}} \nabla_z \log \left( \frac{1}{k} \sum_{i=1}^{k} \frac{p_\epsilon(\epsilon_i) q_{z|\epsilon}(z|\epsilon_i)}{q_{\epsilon|\widetilde{z}}(\epsilon_i|\widetilde{z})} \right) \Bigg|_{\widetilde{z}=z} \quad (16)$$
$$= \nabla_z \log q_z(z).$$

We prove Proposition 3.1 in Section 3.1. Hence, we propose to learn $\tau_{\epsilon|z}$ by minimizing the expected forward Kullback-Leibler divergence $\mathbb{E}_{z \sim q_z} \left[ D_{\mathrm{KL}}(q_{\epsilon|z} \| \tau_{\epsilon|z}) \right]$, for which we can estimate its gradient w.r.t. to the parameters $\theta$ of the NF without bias since

$$\nabla_\theta \mathbb{E}_{z \sim q_z} \left[ D_{\mathrm{KL}}(q_{\epsilon|z} \| \tau_{\epsilon|z}) \right] \quad (17)$$
$$= \mathbb{E}_{z \sim q_z} \mathbb{E}_{\epsilon \sim q_{\epsilon|z}} \nabla_\theta \log \left( \frac{q_{\epsilon|z}(\epsilon|z)}{\tau_{\epsilon|z}(\epsilon|z)} \right) \quad (18)$$
$$= -\mathbb{E}_{z, \epsilon \sim q_{z,\epsilon}} \nabla_\theta \log \tau_{\epsilon|z}(\epsilon|z) \quad (19)$$

which holds because $q_{z,\epsilon}$ does not depend on $\theta$. The following proposition assures the validity of our procedure:

**Proposition 3.2.** *Minimizing $\mathbb{E}_{z \sim q_z} \left[ D_{\mathrm{KL}}(q_{\epsilon|z} \| \tau_{\epsilon|z}) \right]$ is equivalent to minimizing $D_{\mathrm{KL}}(q_{z,\epsilon} \| \tau_{\epsilon|z} \cdot q_z)$.*

This follows from the fact that

$$\mathbb{E}_{z \sim q_z} \left[ D_{\mathrm{KL}}(q_{\epsilon|z} \| \tau_{\epsilon|z}) \right] \quad (20)$$
$$= \mathbb{E}_{z \sim q_z} \mathbb{E}_{\epsilon \sim q_{\epsilon|z}} \log \left( \frac{q_{\epsilon|z}(\epsilon|z)}{\tau_{\epsilon|z}(\epsilon|z)} \right) \quad (21)$$
$$= \mathbb{E}_{z, \epsilon \sim q_{z,\epsilon}} \log \left( \frac{q_{z,\epsilon}(z, \epsilon)}{\tau_{\epsilon|z}(\epsilon|z) q_z(z)} \right) \quad (22)$$
$$= D_{\mathrm{KL}}(q_{z,\epsilon} \| \tau_{\epsilon|z} \cdot q_z) \quad (23)$$

From this, assuming that $\tau_{\epsilon|z}$ is sufficiently flexible, it directly follows that at the global optimum $\tau_{\epsilon|z}^*$ of the expected forward $D_{\mathrm{KL}}$ it holds that

$$q_{z,\epsilon} = \tau_{\epsilon|z}^* \cdot q_z \Rightarrow \tau_{\epsilon|z}^* = \frac{q_{z,\epsilon}}{q_z} = q_{\epsilon|z}. \quad (24)$$

Being of particular importance for understanding our finding, we also include the proof of Proposition 3.1 in the following.

### 3.1. Proof of Proposition 3.1

First note that

$$\frac{p_\epsilon(\epsilon_i)}{q_{\epsilon|z}(\epsilon_i|z)} = \frac{p_\epsilon(\epsilon_i) q_z(z)}{p_\epsilon(\epsilon_i) q_{z|\epsilon}(z|\epsilon_i)} = \frac{q_z(z)}{q_{z|\epsilon}(z|\epsilon_i)}. \quad (25)$$

With this, we get that

$$\mathbb{E}_{\epsilon_i \sim q_{\epsilon|z}} \nabla_z \log \left( \frac{1}{k} \sum_{i=1}^{k} \frac{p_\epsilon(\epsilon_i) q_{z|\epsilon}(z|\epsilon_i)}{q_{\epsilon|\widetilde{z}}(\epsilon_i|\widetilde{z})} \right) \Bigg|_{\widetilde{z}=z} \quad (26)$$
$$= \mathbb{E}_{\epsilon_i \sim q_{\epsilon|z}} \left[ \frac{\frac{1}{k} \sum_{i=1}^{k} \frac{p_\epsilon(\epsilon_i)}{q_{\epsilon|z}(\epsilon_i|z)} \nabla_z q_{z|\epsilon}(z|\epsilon_i)}{\frac{1}{k} \sum_{i=1}^{k} \frac{p_\epsilon(\epsilon_i)}{q_{\epsilon|z}(\epsilon_i|z)} q_{z|\epsilon}(z|\epsilon_i)} \right] \quad (27)$$
$$= \frac{\frac{1}{k} \sum_{i=1}^{k} \mathbb{E}_{\epsilon_i \sim q_{\epsilon|z}} \left[ \frac{p_\epsilon(\epsilon_i)}{q_{\epsilon|z}(\epsilon_i|z)} \nabla_z q_{z|\epsilon}(z|\epsilon_i) \right]}{1/k \sum_{i=1}^{k} q_z(z)} \quad (28)$$
$$= \frac{\sum_{i=1}^{k} \mathbb{E}_{\epsilon_i \sim q_{\epsilon|z}} \left[ \frac{q_z(z) q_{z|\epsilon}(z|\epsilon_i)}{q_{z|\epsilon}(z|\epsilon_i)} \nabla_z \log q_{z|\epsilon}(z|\epsilon_i) \right]}{k \cdot q_z(z)} \quad (29)$$
$$= \frac{1}{k} \sum_{i=1}^{k} \mathbb{E}_{\epsilon_i \sim q_{\epsilon|z}} \left[ \nabla_z \log q_{z|\epsilon}(z|\epsilon_i) \right] \quad (30)$$
$$\overset{\mathrm{Eq.\ 9}}{=} \nabla_z \log q_z(z). \quad (31)$$

### 3.2. Training Under Memory Constraints

One of the main advantages of our proposed score gradient estimators $s_{\mathrm{MC},k}$ and $s_{\mathrm{IS},k}$ is that increasing $k$, i.e., the number of samples $\epsilon_i$, does not increase the computational cost of backpropagation w.r.t. the parameters of our SIVI model $\phi$ because we follow the path gradient. This insight motivates the following procedure, which allows us to train

our SIVI models with constant memory requirements independent of $k$.

First, note that both our score gradient estimators can be written s.t.

$$s(\boldsymbol{z}) = \nabla_{\boldsymbol{z}}\ell(\boldsymbol{z}, \widetilde{\boldsymbol{z}})\bigg|_{\widetilde{\boldsymbol{z}}=\boldsymbol{z}} \quad \text{with} \tag{32}$$

$$\ell(\boldsymbol{z}, \widetilde{\boldsymbol{z}}) = \log\left(\frac{1}{k}\sum_{i=1}^{k} w(\boldsymbol{\epsilon}_i|\widetilde{\boldsymbol{z}})q_{\boldsymbol{z}|\boldsymbol{\epsilon}}(\boldsymbol{z}|\boldsymbol{\epsilon}_i)\right), \tag{33}$$

where choosing $w(\boldsymbol{\epsilon}_i|\widetilde{\boldsymbol{z}}) = 1$ or $w(\boldsymbol{\epsilon}_i|\widetilde{\boldsymbol{z}}) = \frac{p_{\boldsymbol{\epsilon}}(\boldsymbol{\epsilon}_i)}{q_{\boldsymbol{\epsilon}|\widetilde{\boldsymbol{z}}}(\boldsymbol{\epsilon}_i|\widetilde{\boldsymbol{z}})}$ results in $s_{\mathrm{MC},k}$ and $s_{\mathrm{IS},k}$, respectively. Since evaluating $q_{\boldsymbol{\epsilon}|\widetilde{\boldsymbol{z}}}(\boldsymbol{\epsilon}_i|\widetilde{\boldsymbol{z}})$ is computationally non-intensive because of the inner neural sampler, we could, in principle, process very large $\boldsymbol{\epsilon}$ batches. However, since our memory is constrained, we need a way to aggregate score gradient estimators computed on different $\boldsymbol{\epsilon}$ batches.

**Efficient Aggregation on Batch Level** Assume we have computed the score gradient estimates $s_1, s_2$ with associated log probability density estimates $\ell_1, \ell_2$ of the $\boldsymbol{\epsilon}_i$ batches of sizes $j \cdot b$ and $b$, respectively, with $j, b \in \mathbb{N}$. Then, we show in Appendix A.3 that if we aggregate these estimates s.t.

$$\ell_3(\boldsymbol{z}, \widetilde{\boldsymbol{z}}) = \mathrm{logaddexp}\left(\ell_1(\boldsymbol{z}, \widetilde{\boldsymbol{z}}) + \log j, \ell_2(\boldsymbol{z}, \widetilde{\boldsymbol{z}})\right) \\ - \log(j+1), \tag{34}$$

and

$$s_3(\boldsymbol{z}) = \alpha_1 s_1(\boldsymbol{z}) + \alpha_2 s_2(\boldsymbol{z}) \quad \text{with}$$
$$\alpha_1 = \exp\left(\ell_1(\boldsymbol{z}, \widetilde{\boldsymbol{z}}) - \ell_3(\boldsymbol{z}, \widetilde{\boldsymbol{z}}) + \log\frac{j}{j+1}\right), \tag{35}$$
$$\alpha_2 = \exp\left(\ell_2(\boldsymbol{z}, \widetilde{\boldsymbol{z}}) - \ell_3(\boldsymbol{z}, \widetilde{\boldsymbol{z}}) - \log(j+1)\right)$$

then $s_3$ and $\ell_3$ are the corresponding estimates of the combined $\boldsymbol{\epsilon}_i$ batches.

Also, note that we keep most of our operations in the log space to make the procedure numerically stable. For example, we use the $\mathrm{logaddexp}(\ell_1, \ell_2)$ operation, which allows to numerically stable compute $\log(\exp(\ell_1) + \exp(\ell_2))$, and the $\mathrm{logsumexp}$ trick to compute $\ell_1$ and $\ell_2$ themselves. Applying this algorithm iteratively allows us to process an arbitrarily large number of samples $\boldsymbol{\epsilon}_i$ while keeping the memory requirement constant. As a direct consequence, we note that our score gradient estimation is completely parallelizable.

### 3.3. Algorithms

Following the previous findings, we propose two new algorithms for SIVI.

---

**Algorithm 1** BSIVI

---

**Input:** target density $p_{\boldsymbol{z}}$, batch size $m$, number of latent samples $k$ with $k > m$, SIVI model $h_{\boldsymbol{\phi}}$
$i = 1, \ldots, m, \quad j = 1, \ldots, k$
**repeat**
    $\boldsymbol{\epsilon}_j \sim p_{\boldsymbol{\epsilon}}, \boldsymbol{\eta}_j \sim p_{\boldsymbol{\eta}}$
    $\boldsymbol{z}_i = h_{\boldsymbol{\phi}}(\boldsymbol{\epsilon}_i, \boldsymbol{\eta}_i)$
    $s_i = \nabla_{\boldsymbol{z}_i}\mathrm{logsumexp}\left(\left\{\log q_{\boldsymbol{z}|\boldsymbol{\epsilon}}(\boldsymbol{z}_i|\boldsymbol{\epsilon}_j)\right\}_{j=1,\ldots,k}\right)$
    $q_i = \mathrm{stop\_gradient}(s_i) \cdot \boldsymbol{z}_i$
    $\mathrm{loss} = 1/m \sum_{i=1}^{m}(q_i - \log p_{\boldsymbol{z}}(\boldsymbol{z}_i))$
    $\boldsymbol{\phi} = \mathrm{opt}(\mathrm{loss}, \boldsymbol{\phi})$
**until** $\boldsymbol{\phi}$ has converged

---

#### 3.3.1. BSIVI

As a new baseline method, we propose base SIVI (BSIVI), which minimizes the reverse Kullback-Leibler divergence $D_{\mathrm{KL}}(q_{\boldsymbol{z}}\|p_{\boldsymbol{z}})$ by following the path gradient of Eq. 7. For the score gradient $\nabla_{\boldsymbol{z}}\log q_{\boldsymbol{z}}$ we plug-in $s_{\mathrm{MC},k}(\boldsymbol{z})$. This method exploits the fact that we can rapidly sample from a SIVI model, and $s_{\mathrm{MC},k}$ can be computed with constant memory independent of $k$ as discussed in Section 3.2. The algorithm is summarized in Algorithm 1. We use BSIVI to ablate the use of importance sampling, which our main method is built upon.

#### 3.3.2. AISIVI

Furthermore, we propose adaptively informed SIVI (AISIVI), which alternates between minimizing the expected forward KL divergence $E_{\boldsymbol{z}\sim q_{\boldsymbol{z}}}\left[D_{\mathrm{KL}}(q_{\boldsymbol{\epsilon}|\boldsymbol{z}}\|\tau_{\boldsymbol{\epsilon}|\boldsymbol{z}})\right]$ and the reverse KL divergence $D_{\mathrm{KL}}(q_{\boldsymbol{z}}\|p_{\boldsymbol{z}})$ by following the path gradient of Eq. 7. For the score gradient $\nabla_{\boldsymbol{z}}\log q_{\boldsymbol{z}}$, we plug-in $s_{\mathrm{IS},k}(\boldsymbol{z})$, which uses $\tau_{\boldsymbol{\epsilon}|\boldsymbol{z}}$ as the proposal distribution. This alternating training is possible since $s_{\mathrm{IS},k}(\boldsymbol{z})$ is a consistent estimator of the score gradient for any $\tau_{\boldsymbol{\epsilon}|\boldsymbol{z}}$ with $\mathrm{supp}(q_{\boldsymbol{\epsilon}|\boldsymbol{z}}) \subset \mathrm{supp}(\tau_{\boldsymbol{\epsilon}|\boldsymbol{z}})$. Since the forward $D_{\mathrm{KL}}$ is mass covering, we can expect that the support assumption is always fulfilled. This means, in contrast to UIVI, we do not need exact[4] samples from $q_{\boldsymbol{\epsilon}|\boldsymbol{z}}$ and the bias and variance of our estimate decreases[5] with increasing $k$. Also, sampling from the CNF $\tau_{\boldsymbol{\epsilon}|\boldsymbol{z}}$ is comparatively cheap, and the samples are guaranteed to be independent.

## 4. Related Literature

Yin & Zhou (2018) propose to use semi-implicit distributions for VI and train their models by sandwiching the ELBO. Titsias & Ruiz (2019) introduce another objective based on ELBO and derive an associated unbiased gradient

---

[4]However, we can greatly reduce the bias the better we match $q_{\boldsymbol{\epsilon}|\boldsymbol{z}}$ with $\tau_{\boldsymbol{\epsilon}|\boldsymbol{z}}$
[5]This is not the case for UIVI regarding the number of chains

**Algorithm 2** AISIVI

> **Input:** target density $p_{\boldsymbol{z}}$, batch size $m$, number of latent samples $k$, SIVI model $h_{\boldsymbol{\phi}}$, CNF $\tau_{\boldsymbol{\epsilon}|\boldsymbol{z}}$
> $i = 1, \dots, m, \quad j = 1, \dots, k$
> **repeat**
> $\quad \boldsymbol{\epsilon}_i \sim p_{\boldsymbol{\epsilon}}, \boldsymbol{\eta}_i \sim p_{\boldsymbol{\eta}}$
> $\quad \boldsymbol{z}_i = h_{\boldsymbol{\phi}}(\boldsymbol{\epsilon}_i, \boldsymbol{\eta}_i)$
> $\quad \text{loss}_{\text{flow}} = -1/m \sum_{i=1}^{m} \log \tau_{\boldsymbol{\epsilon}|\boldsymbol{z}}(\boldsymbol{\epsilon}_i | \boldsymbol{z}_i)$
> $\quad \boldsymbol{\theta} = \texttt{opt}(\text{loss}_{\text{flow}}, \boldsymbol{\theta})$
>
> $\quad \boldsymbol{\epsilon}_{i,j} \sim \tau_{\boldsymbol{\epsilon}|\boldsymbol{z}}(\cdot | \boldsymbol{z}_i)$
> $\quad \log w_{i,j} = \log p_{\boldsymbol{\epsilon}}(\boldsymbol{\epsilon}_{i,j}) - \log \tau_{\boldsymbol{\epsilon}|\boldsymbol{z}}(\boldsymbol{\epsilon}_{i,j} | \boldsymbol{z}_i)$
> $\quad \log \widetilde{w}_{i,j} = \texttt{stop\_gradient}(\log w_{i,j})$
> $\quad \log \widetilde{q}_{\boldsymbol{z}|\boldsymbol{\epsilon}}(\boldsymbol{z}_i | \boldsymbol{\epsilon}_{i,j}) = \log \widetilde{w}_{i,j} + \log q_{\boldsymbol{z}|\boldsymbol{\epsilon}}(\boldsymbol{z}_i | \boldsymbol{\epsilon}_{i,j})$
> $\quad s_i = \nabla_{\boldsymbol{z}_i} \texttt{logsumexp}\left(\left\{\log \widetilde{q}_{\boldsymbol{z}|\boldsymbol{\epsilon}}(\boldsymbol{z}_i | \boldsymbol{\epsilon}_{i,j})\right\}_{j=1,\dots,k}\right)$
> $\quad q_i = \texttt{stop\_gradient}(s_i) \cdot \boldsymbol{z}_i$
> $\quad \text{loss} = 1/m \sum_{i=1}^{m}(q_i - \log p_{\boldsymbol{z}}(\boldsymbol{z}_i))$
> $\quad \boldsymbol{\phi} = \texttt{opt}(\text{loss}, \boldsymbol{\phi})$
> **until** $\phi$ has converged

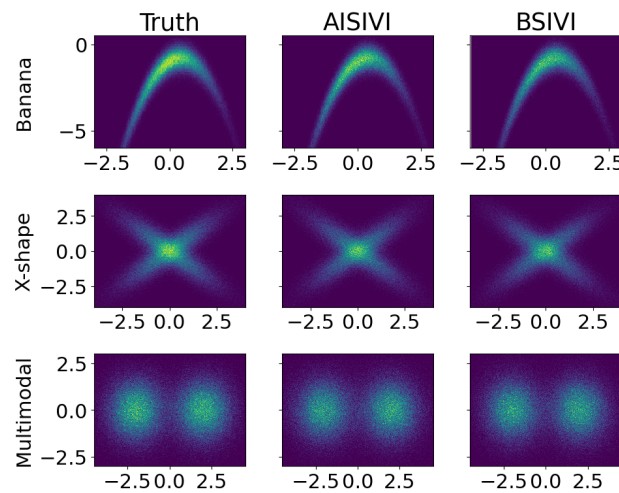

*Figure 2.* Histograms based on 100000 samples produced by the true distribution, AISIVI, and BSIVI

estimator, which, however, depends on expensive MCMC simulations. Sobolev & Vetrov (2019) also improved upon Yin & Zhou (2018) by introducing an importance sampling distribution; however, using expressive models such as CNFs remains infeasible for their approach. In recent years, new approaches based on different objectives have been proposed that seem to outperform methods based on the ELBO. Yu & Zhang (2023) propose minimizing the Fisher divergence, but their minimax formulation proves difficult to train compared to the standard minimization problems mentioned above.

Building upon Yu & Zhang (2023), Cheng et al. (2024) use the kernel Stein discrepancy as the training objective, which turns the minimax problem into a standard minimization problem. We will refer to their method as KSIVI. Lim & Johansen (2024) proposed Particle Semi-Implicit Variational Inference (PVI), which is a particle approximation of a Euclidean-Wasserstein gradient flow. Both Cheng et al. (2024) and Lim & Johansen (2024) showed strong empirical evidence supporting their methods.

While beyond the scope of this work, we note that SIVI has been successfully extended to multilayer architectures, yielding improved performance as demonstrated by Yu et al. (2023).

Beyond SIVI, another line of research explores variational inference with fully implicit distributions (Mescheder et al., 2017; Shi et al., 2018; Feng et al., 2017). These methods often encounter training challenges, such as instability introduced by adversarial learning or density-ratio estimation.

Another related direction performs inference directly in

function space (Sun et al., 2019; Ma et al., 2019; Pielok et al., 2023). These approaches frequently incorporate implicit inference mechanisms within their frameworks.

Several approaches have improved variational inference by incorporating importance sampling. IWAE (Burda et al., 2016) introduces a tighter bound through multiple importance-weighted samples, while NVI (Zimmermann et al., 2021) extends this idea using nested objectives to learn better proposal distributions. Our work builds on this line by integrating importance sampling into the SIVI framework to improve expressivity and stability.

## 5. Experiments

In the following, we analyze the performance of our proposed methods AISIVI and BSIVI under different data scenarios. We start by comparing our two methods on well-known toy examples that serve as a first sanity check (Section 5.1). We then compare our methods with the state-of-the-art methods KSIVI and PVI on a 22-dimensional problem in the context of a Bayesian logistic regression model (Section 5.2, which serves as another common benchmark example for SIVI. Finally, we move to a 100-dimensional problem related to a conditioned diffusion process (Section 5.3). We implemented AISIVI and BSIVI in PyTorch (Paszke et al., 2019). All experiments are performed on a Linux-based server A5000 server with 2 GPUs, 24GB VRAM, and Intel Xeon Gold 5315Y processor with 3.20 GHz.

*Table 1.* $D_{\mathrm{KL}}(p, q)$ of different toy examples (rows) using the two proposed methods (columns).

| NAME | $\downarrow$ AISIVI ($D_{\mathrm{KL}}$) | $\downarrow$ BSIVI ($D_{\mathrm{KL}}$) |
|---|---|---|
| BANANA | 0.0853 | 0.3022 |
| MULTIMODAL | 0.0044 | 0.0017 |
| X-SHAPE | 0.0072 | 0.0034 |

### 5.1. Toy examples

First, we train BSIVI and AISIVI on the three common two-dimensional test densities Banana, X-Shape, and Multimodal as proposed by Cheng et al. (2024). Their respective definitions can be found in Table 3 in the Appendix B. For both methods, we use the same NN architecture and train them for 4000 iterations. For the NF of AISIVI, we use 6 conditional affine coupling layers.

**Results** It can be seen in Figure 2 that AISIVI and BSIVI can capture the three densities nearly equally well. Only for the Banana benchmark, AISIVI outperforms BSIVI notably (Table 1).

### 5.2. Bayesian Logistic Regression

Next, we perform a Bayesian logistic regression on the WAVEFORM[6] dataset as proposed by Yin & Zhou (2018). For the target variables $y_i \in \{0, 1\}, i = 1, \ldots, N$ with $N = 400$ and the feature vectors $\boldsymbol{x}_i \in \mathbb{R}^{21}$, the log-likelihood is given by

$$\log p(y_{1,\ldots,N} \,|\, \boldsymbol{x}_{1,\ldots,N}, \boldsymbol{\beta}) =$$
$$\sum_{i=1}^{N} y_i (1, \boldsymbol{x}_i^\top) \boldsymbol{\beta} - \log\left(1 + \exp\left((1, \boldsymbol{x}_i^\top)\boldsymbol{\beta}\right)\right),$$

where $\boldsymbol{\beta} \in \mathbb{R}^{22}$ is the variable we want to infer. We set the prior distribution of $\boldsymbol{\beta}$ to a normal distribution, i.e., $p(\boldsymbol{\beta}) = \mathcal{N}(0, \alpha^{-1}I)$ with $\alpha = 0.01$. In line with Cheng et al. (2024), we estimate the ground truth by simulating parallel stochastic gradient Langevin dynamics (SGLD Welling & Teh, 2011) for 400,000 iterations, 1000 samples, and a step size of 0.0001. We use the same NN architecture for all methods and use the best hyperparameters for PVI and KSIVI proposed by the respective authors for this benchmark. We train AISIVI and BSIVI for 10,000 iterations and use $\boldsymbol{\epsilon}_i$ batch sizes of 9182 and 91,820 respectively. The large batch size of BSIVI is possible and computationally feasible because of the considerations discussed in Section 3.2. All methods use a batch size $m = 128$ the latent dimension is set to 10, i.e., $\boldsymbol{\epsilon} \in \mathbb{R}^{10}$. For the NF of AISIVI, we use 16 conditional affine coupling layers. We use the

---

[6] https://archive.ics.uci.edu/ml/machine-learningdatabases/waveform

*Table 2.* KSIVI serves as the gold standard, with AISIVI reaching it in 10K iterations. The other SIVI variants are compared based on their estimated log marginal likelihood, given a comparable computational budget to AISIVI. The log marginal likelihood is estimated using 1000 high-quality SGLD samples, while each variant's estimate is computed using 60,000 samples,

| METHOD | $\uparrow$ LOG ML | TRAINING TIME [S] | ITERATIONS |
|---|---|---|---|
| KSIVI | 74521 | 0.6K | 100K |
| AISIVI | **74062** | 1.4K | 10K |
| IWHVI | 67667 | 1.5K | 10K |
| BSIVI | 60556 | 1.5K | 10K |
| PVI | 53121 | 1.4K | 10K |
| UIVI | 40207 | 1.5K | 10K |

full batch for the score gradient computation of the target density.

**Results** The marginal and pairwise density estimates in Figure 3 highlight that all methods perform nearly equally well since no systematic over- or underestimation of the variance can be observed. We also compare with the ground truth all pairwise correlation coefficients of $\boldsymbol{\beta}$ given by

$$\rho_{i,j} = \frac{\mathrm{cov}\left(\boldsymbol{\beta}_{(i)}, \boldsymbol{\beta}_{(j)}\right)}{\sqrt{\mathrm{cov}\left(\boldsymbol{\beta}_{(i)}, \boldsymbol{\beta}_{(i)}\right) \mathrm{cov}\left(\boldsymbol{\beta}_{(j)}, \boldsymbol{\beta}_{(j)}\right)}}, i \neq j, \quad (36)$$

where $\boldsymbol{\beta}_{(i)}$ is a vector containing the $i$-th coordinate of all $\boldsymbol{\beta}$ samples.

The scatter plot in Figure 4 provides a visual summary of the correlation coefficients and the relation between those of different IVI methods and the ones of SGLD as considered ground truth. The results illustrate that PVI and KSIVI exhibit a slightly reduced spread compared to our proposed methods, indicating a marginally better fit. However, overall, the performance of all methods remains comparable.

### 5.3. Conditioned Diffusion Process

We adopt the Bayesian inference setting proposed in Cheng et al. (2024), which is based on the Langevin stochastic differential equation (SDE):

$$dx_t = 10x_t(1 - x_t^2)dt + dw_t, \quad 0 \leq t \leq 1, \quad (37)$$

where $x_0 = 0$ and $w_t$ is a one-dimensional standard Brownian motion. This SDE models the motion of a particle in an energy potential with Brownian fluctuations (Detommaso et al., 2018).

Following (Cheng et al., 2024), we discretize the SDE using the Euler-Maruyama scheme with a step size $\Delta t = 0.01$, yielding a 100-dimensional latent variable

$$\boldsymbol{x} = (x_{\Delta t}, x_{2\Delta t}, \ldots, x_{100\Delta t}),$$

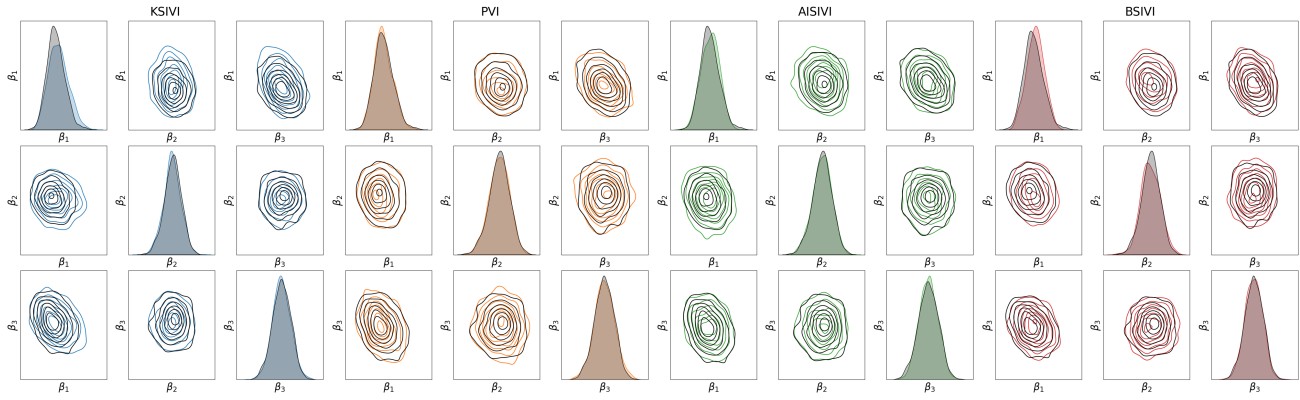

Figure 3. Comparison of marginal and pairwise density estimates of $\boldsymbol{\beta}_{(1)}, \boldsymbol{\beta}_{(2)}, \boldsymbol{\beta}_{(3)}$ where the SGLD estimates are marked in black

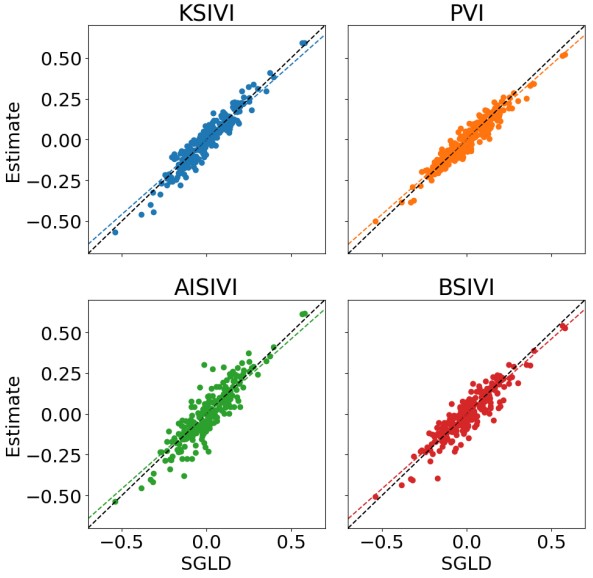

Figure 4. Scatter plot of every pairwise correlation coefficient $\rho_{i,j}$ between the estimates and SGLD.

which gives rise to the prior distribution $p_{\text{prior}}(\boldsymbol{x})$. The observations are perturbed at 20 time points, given by

$$\boldsymbol{y} = (y_{5\Delta t}, y_{10\Delta t}, \ldots, y_{100\Delta t}),$$

where

$$y_{5k\Delta t} \sim \mathcal{N}(x_{5k\Delta t}, \sigma^2), \quad 1 \le k \le 20 \qquad (38)$$

with $\sigma = 0.1$, defining the likelihood function $p(\boldsymbol{y}|\boldsymbol{x})$. Given $\boldsymbol{y}$, our goal is to infer the posterior

$$p(\boldsymbol{x}|\boldsymbol{y}) \propto p_{\text{prior}}(\boldsymbol{x})p(\boldsymbol{y}|\boldsymbol{x}). \qquad (39)$$

To approximate the posterior, we reapply the approach in (Cheng et al., 2024) by running a long-run parallel stochastic gradient Langevin dynamics (SGLD) simulation with 1000 independent particles, a step size of $0.0001$, and $100,000$ iterations to generate 1000 ground truth samples.

For this benchmark, we also include IWHI to ablate the effect of their joint training approach compared to our sequential training. For their method, we use a conditional Gaussian model, where the conditional parameters are predicted by a neural network, as their joint training setup makes more complex conditional models—such as continuous normalizing flows (CNFs)—infeasible. Additionally, we evaluate against UIVI to compare our importance sampling-based enhancement with their original MCMC-based approach. For all methods, we use the same NN architecture. For KSIVI, we use the hyperparameters proposed by the authors for this benchmark. For PVI, we use $100$ particles. To ensure a fair comparison, we fixed the outer batch size (number of sampled $\boldsymbol{z}$) for all SIVI methods and adjusted the inner batch size (number of sampled $\boldsymbol{\epsilon}$) until we achieved approximately the same iterations per second as AISIVI. The $\boldsymbol{\epsilon}_i$ batch sizes for AISIVI, BSIVI, and IWHI are 256, 40960, and 7000, respectively. The latent dimension is $100$ for all SIVI variants. For the NF of AISIVI, we use 32 conditional affine coupling layers.

**Results** The results of the experiment is depicted in Figure 5. We observe that KSIVI and AISIVI are closest to SGLD while UIVI, PVI, and BSIVI tend to underestimate the variability of the process. In Table 2, we report the estimated log marginal likelihoods of the SIVI variants along with their associated training times. Notably, only our method, AISIVI, approaches the performance of the state-of-the-art KSIVI. While IWHI also performs well, it does not match AISIVI, highlighting the benefits of a more expressive proposal model. For UIVI, we were limited to an inner batch size of 2 due to computational constraints, which led to noticeably weaker performance. Nevertheless, this

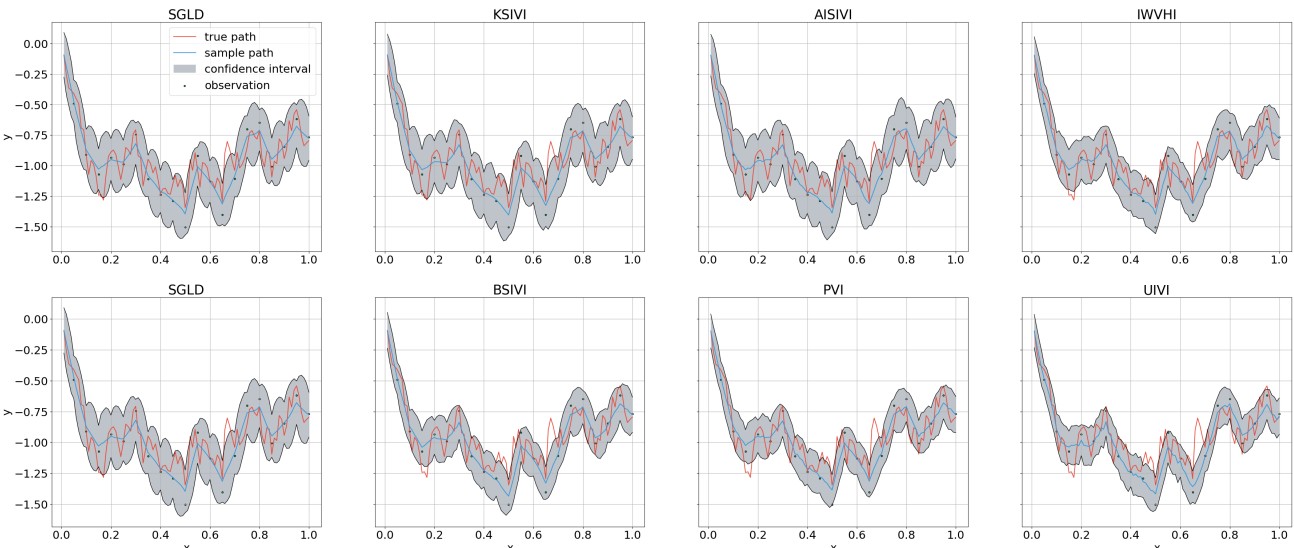

Figure 5. Approximations of KSIVI, PVI, IWHVI, UIVI, AISIVI, and BSIVI for the discretized conditioned diffusion process are shown. The red dots represent the observations, the magenta line the ground truth estimated via parallel SGLD, and the blue line the estimated posterior mean. The shaded region shows the 95 marginal posterior confidence interval at each discretization step

comparison shows that AISIVI successfully adapts UIVI's core ideas in a way that makes them more computationally efficient and competitive.

## 6. Conclusion

In this paper, we proposed a novel SIVI framework, AISIVI, which revitalizes the ELBO as the training objective. This is possible because the bias and variance of the ELBO gradients can be severely reduced by using importance sampling and the optimal proposal distribution can be stably learned with a CNF. We provided the respective efficient Monte Carlo gradient estimators. The numerical experiments support the efficiency and effectiveness claim of AISIVI.

In particular, our experiments on the high-dimensional diffusion example suggest that it can be beneficial not to rely on a kernel method, which is known to be scalable to very large dimensions. Our method thus represents an easy-to-use and scalable alternative to current state-of-the-art SIVI methods with on-par performance.

### Limitations and Outlook

This work marks an initial attempt to integrate the strengths of semi-implicit distributions and normalizing flows. However, given the numerous normalizing flow frameworks, certain alternative combinations may lead to improved performance. Future research could explore these possibilities to identify more effective configurations. While our method shows on par performance with current state-of-the-art SIVI

methods, a suitable combination could further notably enhance performance.

Additionally, the proposed method does not inherently offer exploration capabilities, which may limit its ability to model multi-modal distributions. However, note that we can always combine a temperature annealing strategy (Rezende & Mohamed, 2015) with our approach, but a more principled procedure would be desirable. While this limitation is common in related work, addressing it in future research could enhance the applicability of AISIVI.

## Impact Statement

This paper presents work whose goal is to advance the field of machine learning. There are many potential societal consequences of our work, none of which we feel must be specifically highlighted here.

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

# A. Proofs

For the proofs, we assume that the objects of interest are sufficiently regular s.t. we can change the order of integration, summation, and differentiation.

## A.1. $s_{\mathrm{MC},k}$ is a consistent estimator and its bias approximation

We approximate the bias of $s_{\mathrm{MC},k}(z) = \nabla_z \log\left(\frac{1}{k}\sum_{i=1}^k q_{z|\epsilon}(z|\epsilon_i)\right)$ by using the delta method. First we note for

large $k$ that $s_{\mathrm{MC},k}(z) \approx \underbrace{\nabla_z \log\left(\frac{1}{k-1}\sum_{i=2}^k q_{z|\epsilon}(z|\epsilon_i)\right)}_{=:\widetilde{s_{\mathrm{MC},k}}(z)}$. With the second-order Taylor approximation around $q_z(z) =$

$\mathbb{E}_{\epsilon_i \sim p_\epsilon}\left[\frac{1}{k}\sum_{i=2}^{k-1} q_{z|\epsilon}(z|\epsilon_i)\right]$ we get that

$$\log\left(\frac{1}{k-1}\sum_{i=2}^k q_{z|\epsilon}(z|\epsilon_i)\right) \approx \log\left(q_z(z)\right) + \frac{\frac{1}{k-1}\sum_{i=2}^k q_{z|\epsilon}(z|\epsilon_i) - q_z(z)}{q_z(z)} - \frac{\left(\frac{1}{k-1}\sum_{i=2}^k q_{z|\epsilon}(z|\epsilon_i) - q_z(z)\right)^2}{2\cdot q_z(z)^2}. \quad (40)$$

From this, it follows that

$$\mathbb{E}_{\epsilon_i \sim p_\epsilon}\left[\log\left(\frac{1}{k-1}\sum_{i=2}^k q_{z|\epsilon}(z|\epsilon_i)\right)\right] \approx \log\left(q_z(z)\right) - \frac{\mathbb{V}_{\epsilon\sim p_\epsilon}\left[q_{z|\epsilon}(z|\epsilon)\right]}{2(k-1)\cdot q_z(z)^2}. \quad (41)$$

Consequently, we get for large $k$ that

$$\mathbb{E}_{\epsilon_i \sim p_\epsilon}\left[\nabla_z s_{\mathrm{MC},k}(z)\right] - \nabla_z \log\left(q_z(z)\right) \approx \mathbb{E}_{\epsilon_i \sim p_\epsilon}\left[\nabla_z \widetilde{s_{\mathrm{MC},k}}(z)\right] - \nabla_z \log\left(q_z(z)\right) \quad (42)$$

$$\approx -\nabla_z\left(\frac{\mathbb{V}_{\epsilon\sim p_\epsilon}\left[q_{z|\epsilon}(z|\epsilon)\right]}{2(k-1)\cdot q_z(z)^2}\right), \quad (43)$$

which, in general, is non-zero.

To prove the consistency of $s_{\mathrm{MC},k}$, we observe since $\log$ is a continuous function that

$$\lim_{k\to\infty} s_{\mathrm{MC},k} = \nabla_z \log\left(\lim_{k\to\infty}\frac{1}{k}\sum_{i=1}^k q_{z|\epsilon}(z|\epsilon_i)\right) \overset{\mathrm{a.s.}}{=} \nabla_z \log\left(\mathbb{E}_{\epsilon\sim p_\epsilon}\left[q_{z|\epsilon}(z|\epsilon)\right]\right) = \nabla_z \log\left(q_z(z)\right), \quad (44)$$

i.e.,

$$\mathbb{P}\left(\lim_{k\to\infty} s_{\mathrm{MC},k} = \nabla_z \log\left(q_z(z)\right)\right) = 1. \quad (45)$$

## A.2. $s_{\mathrm{IS},k}$ is a consistent estimator

To prove the consistency of $s_{\mathrm{IS},k}$, we observe since $\log$ is a continuous function that

$$\lim_{k\to\infty} s_{\mathrm{IS},k} = \nabla_z \log\left(\lim_{k\to\infty}\frac{1}{k}\sum_{i=1}^k \frac{p_\epsilon(\epsilon)q_{z|\epsilon}(z|\epsilon_i)}{\tau_{\epsilon|\widetilde{z}}(\epsilon_i|\widetilde{z})}\right)\Bigg|_{\widetilde{z}=z} \overset{\mathrm{a.s.}}{=} \nabla_z \log\left(\mathbb{E}_{\epsilon_i\sim\tau_{\epsilon|z}}\left[\frac{p_\epsilon(\epsilon)q_{z|\epsilon}(z|\epsilon_i)}{\tau_{\epsilon|\widetilde{z}}(\epsilon_i|\widetilde{z})}\right]\right)\Bigg|_{\widetilde{z}=z}. \quad (46)$$

For a valid proposal distribution, it must hold that $\tau_{\epsilon|z}$ must be non-zero where $p_\epsilon \cdot q_{z|\epsilon} = q_{z,\epsilon}$ is greater than zero (Owen, 2013). Consequently, the support of $\tau_{\epsilon|z}$ must also contain the support of $q_{\epsilon|z} = \frac{q_{z,\epsilon}}{q_z}$. In this case

$$\lim_{k\to\infty} s_{\mathrm{IS},k} \overset{\mathrm{a.s.}}{=} \nabla_z \log\left(q_z(z)\right), \text{ i.e., } \mathbb{P}\left(\lim_{k\to\infty} s_{\mathrm{IS},k} = \nabla_z \log\left(q_z(z)\right)\right) = 1. \quad (47)$$

**A.3. $s_3$ gives the correct score gradient estimator regarding all $\epsilon_i$ samples**

Assume we got a $\epsilon$ batch of size $(j + 1) \cdot b$ and have computed the following estimators

$$s_1(z) = \nabla_z \ell_1(z, \widetilde{z})\Big|_{\widetilde{z}=z} \text{ with} \tag{48}$$

$$\ell_1(z, \widetilde{z}) = \log \left( \frac{1}{j \cdot b} \sum_{i=1}^{j \cdot b} w(\epsilon_i|\widetilde{z}) q_{z|\epsilon}(z|\epsilon_i) \right), \tag{49}$$

$$s_2(z) = \nabla_z \ell_2(z, \widetilde{z})\Big|_{\widetilde{z}=z} \text{ with} \tag{50}$$

$$\ell_2(z, \widetilde{z}) = \log \left( \frac{1}{b} \sum_{i=j \cdot b+1}^{(j+1) \cdot b} w(\epsilon_i|\widetilde{z}) q_{z|\epsilon}(z|\epsilon_i) \right). \tag{51}$$

These estimates can be aggregated such that

$$\ell_3(z, \widetilde{z}) = \text{logaddexp}\left(\ell_1(z, \widetilde{z}) + \log j, \ell_2(z, \widetilde{z})\right) - \log(j + 1), \tag{52}$$

$$= \log \left( \frac{1}{(j+1) \cdot b} \sum_{i=1}^{(j+1) \cdot b} w(\epsilon_i|\widetilde{z}) q_{z|\epsilon}(z|\epsilon_i) \right), \tag{53}$$

$$s_3(z) = \alpha_1 s_1(z) + \alpha_2 s_2(z) \text{ with} \tag{54}$$

$$\alpha_1 = \exp \left( \ell_1(z, \widetilde{z}) - \ell_3(z, \widetilde{z}) + \log \frac{j}{j + 1} \right), \tag{55}$$

$$\alpha_2 = \exp \left( \ell_2(z, \widetilde{z}) - \ell_3(z, \widetilde{z}) - \log(j + 1) \right). \tag{56}$$

$$\tag{57}$$

For the score gradient estimate, it follows that

$$s_3 = \frac{1}{\exp(\ell_3(z, \widetilde{z}))} \nabla_z \frac{j}{j + 1} \exp\left(\ell_1(z, \widetilde{z})\right)\Big|_{\widetilde{z}=z} + \frac{1}{\exp(\ell_3(z, \widetilde{z}))} \nabla_z \frac{1}{j + 1} \exp\left(\ell_2(z, \widetilde{z})\right)\Big|_{\widetilde{z}=z} \tag{58}$$

$$= \frac{1}{\exp(\ell_3(z, \widetilde{z}))} \nabla_z \exp\left(\ell_3(z, \widetilde{z})\right)\Big|_{\widetilde{z}=z} \tag{59}$$

$$= \nabla_z \ell_3(z, \widetilde{z})\Big|_{\widetilde{z}=z}. \tag{60}$$

# B. Implementation Details

Table 3 summarizes the details for the toy example discussed in Section 5.1.

*Table 3.* Densities of the toy examples

| NAME | DENSITY | PARAMETERS |
|------|---------|------------|
| BANANA | $z = (\nu_1, \nu_1^2 + \nu_2 + 1)^\top, \nu \sim \mathcal{N}(0, \Sigma)$ | $\Sigma = \begin{bmatrix} 1 & 0.9 \\ 0.9 & 1 \end{bmatrix}$ |
| MULTIMODAL | $z \sim 0.5\mathcal{N}(z\mid \mu_1, I) + 0.5\mathcal{N}(z\mid \mu_2, I)$ | $\mu_1 = (-2, 0)^\top, \mu_2 = (2, 0)^\top$ |
| X-SHAPE | $z \sim 0.5\mathcal{N}(z\mid 0, \Sigma_1) + 0.5\mathcal{N}(z\mid 0, \Sigma_2)$ | $\Sigma_1 = \begin{bmatrix} 2 & 1.8 \\ 1.8 & 2 \end{bmatrix}, \Sigma_2 = \begin{bmatrix} 2 & -1.8 \\ -1.8 & 2 \end{bmatrix}$ |

