# OpenReview forum: "Revisiting Unbiased Implicit Variational Inference"
_ICML.cc/2025/Conference — ICML 2025 poster_

### Official Review · Reviewer_2yqc · 2025-02-25

**Overall Recommendation:** 4

**Summary:**

In this work they propose importance sampling estimation of the score function needed for minimising the KL divergence between q_z and p_z in SIVI. To do this they use a CNF proposal.

They compare their methods (one which uses the importance sampling estimator and one which doesn't) against a kernel stein discrepancy based method (KSIVI) and a particle approximation to a Euclidean-Wasserstein gradient flow (PVI). They show that their methods perform comparably to PVI and KSIVI on a Bayesian Logistic Regression task, and that their importance sampling based method performs favourably (both in terms of approximation quality and clock-time) on a Conditioned Diffusion Process task.

## update after rebuttal
I maintain my recommendation.

**Claims And Evidence:**

I believe the authors show that their proposed method works and is comparable to other existing methods, while saving on computation thanks to the replacement of the costly MCMC step with importance sampling.

**Essential References Not Discussed:**

Not that I'm aware of.

**Experimental Designs Or Analyses:**

As previously mentioned the experiments seemed sensible.

**Methods And Evaluation Criteria:**

Yes, the datasets and models evaluated on seem adequate.

**Other Comments Or Suggestions:**

"principle manner" line 029

From line 55 to 62 in the second column, I wonder if Andrade (2024) might not be a better reference as they have a theorem aimed at this very question, whereas VI: A review for statisticians seems to suggest that underestimation of the variance is "a consequence of its objective function"?

In algorithm 2 it isn't clear what theta is for (I'm guessing the parameters of tau).

**Other Strengths And Weaknesses:**

I thought this was a well written paper, using the background section to both cover the state of the field but also motivate the development of the presented method was a nice touch.

Section 3.2 was interesting and I appreciated that some thought was given to efficient implementation.

In the results I would have liked to have seen iterations per second for all the methods used in the final example.

**Questions For Authors:**

On line 250 in column 2: "This alternating training is possible since sIS,k(z) is a consistent estimator of the score gradient for any τε|z with supp(qε|z) ⊂ supp(τε|z)."

Could you explain this further?

**Relation To Broader Scientific Literature:**

The work builds on Titsias & Ruiz (2019) which introduces Unbiased Implicit Variational Inference, and in particular includes the path gradient estimator which is needed to make their importance weighted estimator of the score function work.

This work also relates to Yin & Zhou (2018) in that they both use a semi-implicit distribution as a proposal.

**Theoretical Claims:**

I looked over the proof in the main text which seemed correct to me. I also looked at some of the proofs of consistency in the appendix and could see no issues.

---

> ### Author Rebuttal · Authors · 2025-04-01
>
> Thank you very much for your constructive feedback. We have made the following revisions in response to your comments:
>
> 1. **Computational Cost Comparison**: Following your suggestion to investigate computational costs, we have now performed comparisons based on the Conditioned Diffusion Process task for all methods. To facilitate a fair comparison, we use the total iterations of KSIVI (outer loop iterations × inner loop iterations) instead of just the outer iterations as done in Sec 5.3. This comparison highlights that AISIVI learns much faster per iteration than KSIVI (as shown in the [table](https://figshare.com/s/f45697639e8908df84dd?file=53348411)), but also emphasizes the extraordinarily fast implementation of KSIVI when comparing run times. Overall, the comparison confirms that only the gold standard KSIVI and our AISIVI reach log marginal likelihood values > 70,000, while all other methods notably lag behind. This result aligns with the visual comparison in [Fig. 4](https://figshare.com/s/f45697639e8908df84dd?file=53348420) and demonstrates that it is possible to achieve such performance without relying on a kernel-based approach.
>
>
> 2. **Explanation for Consistency**: Regarding your question
> > On line 250 in column 2: "This alternating training is possible since sIS,k(z) is a consistent estimator of the score gradient for any τε|z with supp(qε|z) ⊂ supp(τε|z)."
>
> about the consistency of alternating training: This statement means that as long as K (the number of importance samples) is large enough, we can draw from any proposal distribution that satisfies the support assumption. Since the score gradient estimator ($s_{\text{IS}}$) is consistent, it converges almost surely. This allows us to keep the proposal distribution fixed while updating the SIVI model, and update the flow afterwards with our unbiased forward KL gradient, which moves the current CNF closer to the optimal proposal distribution (the target) for the new SIVI model. This enables convergence to the global solution. We will make this more clear in an updated paper version and thank the reviewer for this question.
>
> 3. **Typo and Reference Update**: We have corrected the typo and greatly appreciate the reference to Andrade (2024). We will replace the current citation with Andrade (2024), as it provides a more relevant theorem on variance underestimation.
>
>
> 4. **Algorithm Clarification**: We updated the algorithm description to make it clear that θ refers to the parameters of τ (the normalizing flow). This change should clarify the notation and its role in the model.
>
>
> We hope these revisions address your concerns. Thank you again for your detailed feedback, which has been incredibly helpful in improving the clarity and rigor of our work.

---

> > ### Comment · Reviewer_2yqc · 2025-04-02
> >
> > Thank you for responding to my concerns. In light of the additional experiments and commitment to clarifications in the manuscript, as well as addressing the concerns of the other reviewers I will maintain my recommendation.

---

### Official Review · Reviewer_yGYH · 2025-03-02

**Overall Recommendation:** 4

**Summary:**

This paper proposes a new method to reduce the bias of semi-implicit VI (SIVI). The key idea is  to estimate the problematic term $\nabla_z \log q(z) = \nabla_z \log E_{\epsilon} [q(z|\epsilon) ]$ using importance sampling, where the proposal distribution is a normalizing flow. The normalizing flow is learned to match the posterior distribution $q(\epsilon|z)$. If learned perfectly, it completely debiases the estimator. The authors propose a neat way for how the IS can be summed efficiently for large batches. Several experiments on different inference settings have been performed where the papers that the proposed method can compete with SOTA inference methods.

**Claims And Evidence:**

Yes

**Essential References Not Discussed:**

[1] Ma, Chao, Yingzhen Li, and José Miguel Hernández-Lobato. "Variational implicit processes." International Conference on Machine Learning. PMLR, 2019.

[2] Yu, Longlin, et al. "Hierarchical semi-implicit variational inference with application to diffusion model acceleration." Advances in Neural Information Processing Systems 36 (2023): 49603-49627.

[3] Shi, Jiaxin, Shengyang Sun, and Jun Zhu. "Kernel implicit variational inference." arXiv preprint arXiv:1705.10119 (2017).

[4] Zimmermann, Heiko, et al. "Nested variational inference." Advances in Neural Information Processing Systems 34 (2021): 20423-20435.

**Experimental Designs Or Analyses:**

The experimental design makes sense

**Methods And Evaluation Criteria:**

The evaluation Criteria makes sense

**Other Comments Or Suggestions:**

- In Figure 4, the magenta line and the blue line are not very color blind friendly and are hard to separate I would advise to use different coloring scheme . Also please use legends and label the x and y axis in your figures.

**Other Strengths And Weaknesses:**

***Strengths*

- The paper reads very well. The motivation of the paper makes sense and it offers a nice introduction to SIVI and the problems with it.

- While the proposed approach might be simple, it makes a lota of sense and is a perfectly valid contribution for ICML.

- The experiments are very systematic and well designed, on par with many other VI papers.

**Weaknesses**

- I must say the results seem abit underwhelming. One issue is that ofcourse most of the results figures (2,3,4) are difficult to read and judge, in the sense that is diffcult to compare the performance of a method to the ground truth as well as other methods. There are no tables with the exception of the toy experiment. It looks to me that everything performs roughly the same, while KSVI and PVI perform a bit better. I understand that beating SOTA is not a requirement but it should be explained abit given that these methods are used as baselines that what is the (potential) advantage here? Also how well the spline flow itself would have performed by itself (simply minimizing $KL[q|p]$ where $q$ is the NF.)

- Related to the previous issues, some discussion of speed of convergence, its stability, and hyperparameter sensitivity would have been a good improvement. Having an expectation inside the gradient term (Eq. 14) seems very sensitive. I would expect $K$ to have to be large? How sensitive was the choice of $K$?

**Questions For Authors:**

See weaknesses and Strengths

**Relation To Broader Scientific Literature:**

I must admit I am not very familiar with the literature in this regard. I have listed a few papers that seem relevant which are not cited here.

**Theoretical Claims:**

I only checked 3.1 which to the best of my understanding, is correct.

---

> ### Author Rebuttal · Authors · 2025-04-01
>
> Thank you for your thoughtful feedback and valuable suggestions. We have made several improvements to address your concerns:
>
> 1. **Improved Figures and Tables**: We have revised Figures 2, 3, and 4 for better clarity and ensured they are color-blind-friendly. Specifically, we have revised the [figure of Experiment 3](https://figshare.com/s/f45697639e8908df84dd?file=53348420) to better illustrate the performance differences. Additionally, we have added a [table](https://figshare.com/s/f45697639e8908df84dd?file=53348411) for clearer comparisons. The updated results now show that in Experiment 3, AISIVI and KSIVI outperform all other methods in terms of log marginal likelihood.
>
>
> 2. **Additional Baselines**: We have expanded our comparisons to include UIVI and IWHVI. Furthermore, for the multimodal example, we added a [normalizing flow baseline](https://figshare.com/s/f45697639e8908df84dd?file=53348414). The results demonstrate that normalizing flows either smooth out low-density regions or introduce additional fragments (as observed with the neural spline flow).
>
> 3. **Convergence Speed and Stability**: KSIVI remains the fastest in terms of convergence speed, but our method is the only one that matches its performance. Interestingly, AISIVI does not require very large inner batches if the proposal distribution (e.g., a normalizing flow) is sufficiently flexible and learns faster per iteration than KSIVI. Our framework efficiently supports computationally expensive proposal distributions, such as CNFs, since we do not need to backpropagate through the proposal distribution when computing path gradients. This property is one of our main novel contributions. However, without a proposal distribution, large batch sizes are indeed necessary, as demonstrated by BSIVI.
>
>
> 4. **Computational Cost Comparison**: Following the suggestion to investigate computational costs, we have now performed comparisons based on the Conditioned Diffusion Process task for all methods. To facilitate a fair comparison, we now use the total iterations of KSIVI (outer loop iterations × inner loop iterations) instead of the outer iterations as done in Sec 5.3. While this shows that AISIVI learns much faster per iteration than KSIVI (as seen in the table), it also highlights the extraordinarily fast implementation of KSIVI when comparing run times. Overall, the comparison clearly shows that only the gold standard KSIVI and our AISIVI are able to reach log marginal likelihood values >70,000, while all other methods lag behind significantly. This confirms the visual comparison of Fig. 4 in the paper and demonstrates that such performance levels can be achieved without relying on a kernel-based approach.
>
>
> 5. **Citations**: We will incorporate all the suggested references and are very grateful for your recommendations, which further enhance our paper’s positioning within the broader literature.
>
>
> Thank you again for your constructive feedback, which has helped us refine and strengthen our work. We hope these improvements address your concerns.

---

> > ### Comment · Reviewer_yGYH · 2025-04-02
> >
> > I have read the rebuttal and want to thank the authors for their response.
> >
> > Given the new results, I think the paper is good form, and I'm still happy to recommend acceptance. Regarding the concerns of other reviewers about baselines, it is hard for me to judge because I am not very familiar with this literature so I will ask AC to weigh other reviews more on this front.

---

### Official Review · Reviewer_dhXk · 2025-03-08

**Overall Recommendation:** 3

**Summary:**

Estimating the gradient of the KL divergence between SIVI models and (unnormalized) densities is the core difficulty for training SIVI models. Many efforts have been made to partially solve this problem using, e.g., MCMC, kernel methods, Monte Carlo sampling, etc.
This paper presents a new method for training semi-implicit variational inference (SIVI) models based on importance sampling (IS).
The most contribution of this paper is employing a learnable reverse model $\tau(\epsilon|z)$ to provide a variance-reduced estimate of the gradient.
The experiments show that the proposed method has a comparable inference accuracy as baseline methods.

**Claims And Evidence:**

There are two important claims that are not verified by the experiments.
- **On the high-dimensional application**. Line 57-59 explicitly states that 'In this work, ... enable us to train SIVI models even in high dimensions'. Line 151 states that "they can be scaled up to high dimensions". However, the largest dimension of the numerical examples is 100, which is not so large and is well studied by baseline methods (KSIVI). The authors should provide more high-dimensional evidence (e.g., AISIVI performs well and baseline methods fail) to support their claim.
- **On the computational cost**. The authors emphasized the computational drawback of UIVI in Section 2.3 and "propose a novel method to fix the shortcomings" (Line 124). However, no detailed computational cost comparison, or the efficiency/accuracy trade-off is reported, making their claim unconvincing.

**Essential References Not Discussed:**

No.

**Experimental Designs Or Analyses:**

### **About the experimental designation**
- **Missing high-dimensional application**. Please see 'claims and evidence.'
- **Missing computational cost comparison**. Please see 'claims and evidence.'

### **About the baselines**
As least the following two variants of SIVI should be considered:
- UIVI. This paper "revisits" UIVI (see the title), by using important sampling to "fix the encountered shortcomings of UIVI" (Line 124). Therefore, UIVI should be considered as a very important baseline for the method.
- IWHVI. This paper has the same idea of using importance sampling to estimate the ELBO in optimizing SIVI.

### **About the metrics**
From the figures, AISIVI achieves very indistinguishable results compared to the baseline methods. In such a case, quantitative comparison (e.g., KL divergence, Wasserstein distance, ELBO, marginal likelihood) would be necessary to compare different methods.

**Methods And Evaluation Criteria:**

Although using important sampling to estimate the gradient is a somewhat straightforward, the authors make theoretical analysis and careful designation which can be considered as novel contributions. However, there are some problematic point in the methodology:
- Regarding the comparison against IWHVI (Soblev \& Vetrov, 2019), the author claims IWHVI is not applicable to the reparameterization trick. However, reparameterization trick is possible for IWHVI, as both the SIVI model and the reverse model are reparametrizable. I would like to see more explanation on this point.
- The authors use an alternate optimization strategy for the SIVI model and importance distribution. A more natural choice is to optimize these two models simultaneously, and this strategy has been applied in VAE and IWHVI. It would be better to explain why an alternate optimization is considered here.

**Other Comments Or Suggestions:**

One typo: There should be $\epsilon_i$ in the numerator of Eq. 15.

**Other Strengths And Weaknesses:**

I have no other comments.

**Questions For Authors:**

I have no other questions.

**Relation To Broader Scientific Literature:**

Using importance sampling to improve the gradient estimation of SIVI is a meaningful effort, as this is a long-standing and difficult problem of SIVI.

**Theoretical Claims:**

I've checked the correctness of the theoretical claims.

---

> ### Author Rebuttal · Authors · 2025-04-01
>
> Thank you for your thorough review and insightful feedback. We greatly appreciate your suggestions, and we have addressed the following points:
>
> 1. **High-dimensional Applicability**: We argue that the definition of "high-dimensional" depends on the specific goal. Finding an adequate approximation is achievable in higher dimensions, but our target for Experiment 3 was the true distribution. As shown in the newly added [table](https://figshare.com/s/f45697639e8908df84dd?file=53348411) and improved [figure](https://figshare.com/s/f45697639e8908df84dd?file=53348420), only our method and KSIVI from all SIVI variants come close to matching this distribution. Thus, for this scenario, 100 dimensions can be considered sufficiently high, as no other SIVI variant can achieve such good results. We will clarify in the final manuscript what we mean by 'high-dimensional' in our setting to avoid any confusion.
>
>
> 2. **Computational Cost**: We have now included UIVI in Experiment 3, and as you suggested, we’ve focused on the log marginal likelihood. It is clear that UIVI’s performance does not come close to AISIVI, even for a similar simulation budget. Note that we can only afford an inner batch size of 2 because of the expensive inner MCMC loop, which further emphasizes our point about drastically improved computational costs. Similar findings are reported in Appendix D of SEMI-IMPLICIT VARIATIONAL INFERENCE VIA SCORE MATCHING by Longlin Yu et al., which further supports our argument. Thank you for strengthening our paper with this suggestion.
>
>
> 3. **Reparameterization Trick and IWHVI**: We have corrected our previous claim, and indeed, the reparameterization trick can be applied to IWHVI. We appreciate your clarification on this point.
>
>
> 4. **Optimization Strategy**: The alternate optimization scheme is a key aspect of our method and one of the main contributions. By using the path gradient and alternating optimization, we avoid backpropagating through the proposal distribution. This allows us to process arbitrarily large inner batches, whereas a simultaneous optimization approach is limited by memory constraints. This design enables us to use a more computationally expensive proposal distribution (conditional normalizing flow), and as shown in our updated experiments, AISIVI outperforms IWHVI in terms of performance within a similar computational budget. This now serves as an ablation study comparing simultaneous and alternating training approaches. We have also included IWHVI as a strong baseline, as suggested.
>
>
> 5. **Typo**: We have corrected the typo in Eq. 15, as you pointed out.
>
>
> Thank you again for your valuable feedback and for helping to improve the clarity and strength of our paper. We hope these changes address your concerns effectively. Should the reviewer find the response satisfactory, we would appreciate reconsidering the initial score. Otherwise, we remain fully committed to addressing any remaining concerns during the second author response phase.

---

> > ### Comment · Reviewer_dhXk · 2025-04-03
> >
> > Thank you for the additional results. For table 2 in your rebuttal, I still have some concerns.
> > - On which experiment do you report these marginal likelihoods? and the device type?
> > - As you have argued, IWHVI can be considered as a simultaneous ttaining ablation for AISIVI. Then why IWHVI uses more time than AISIVI?
> > - I suggest the authors to give a more comprehensive study about the training cost, maybe a dimension scaling curve, since they have made strong claims on the reduced computational cost of AISIVI.
> >
> > Regarding the high-dimensional experiments, I still think such a numerical example (e.g., Bayesian neural network) would greatly strengthen the paper. However, it still looks good if other advantages are demonstrated.

---

> > > ### Author Response · Authors · 2025-04-04
> > >
> > > We thank the reviewer for reading our rebuttal and providing additional feedback. We address the mentioned comments in the following point-by-point:
> > >
> > > 1. **Marginal Likelihoods and Device Details**: As mentioned in the previous response, the marginal likelihoods refer to Experiment 3 (the conditioned diffusion process). The updated figures illustrate where other methods fail in contrast to AISIVI and KSIVI. Regarding the device, as stated in the main text, the experiments were conducted on a Linux-based A5000 server with 2 GPUs, 24GB VRAM, and an Intel Xeon Gold 5315Y processor with a clock speed of 3.20 GHz.
> > >
> > >
> > > 2. **Training Time and Batch Size Adjustment**: We thank the reviewer for this valid comment. We, however, think that there is a misunderstanding. In this experiment, we wanted to compare the performance of models when fixing the budget of all methods to address the "efficiency/accuracy trade-off" you requested in your initial review. More specifically, we provided all methods with 10k iterations and approximately the same computational budget as AISIVI, which is reflected in the runtime data. To ensure a fair comparison, we fixed the outer batch size (number of sampled z) for all SIVI methods and adjusted the inner batch size (number of sampled epsilons) until we achieved approximately the same iterations per second as AISIVI. For IWHVI, this resulted in an inner batch size of 7000. As shown in the IWHVI paper, larger inner batch sizes tend to improve performance.
> > >
> > >
> > > 3. **Computational Cost Claims**: Our main claim in the paper is that AISIVI vastly outperforms UIVI in terms of efficiency. This is clearly shown in the updated Experiment 3 (in the rebuttal). In that experiment, we could only afford an inner batch size of 2 for UIVI, resulting in significantly worse performance compared to AISIVI. However, we are happy to provide a dimensional scaling plot in a revised version of the paper to further underpin this claim should the reviewer require further evidence.
> > > We hope these points help to clarify the additional comments by the reviewer.
> > >
> > > Thank you again for your thoughtful feedback! We appreciate the reviewer’s suggestion regarding a high-dimensional example. We recognize its value in further illustrating our findings and are currently exploring the feasibility of incorporating such a case within the scope of this revision.

---

### Official Review · Reviewer_Ssns · 2025-03-14

**Overall Recommendation:** 3

**Summary:**

This paper revisits Unbiased Implicit Variational Inference (UIVI), which has been largely dismissed due to its computational cost and imprecision from the inner MCMC loop. The authors propose replacing MCMC with importance sampling. By minimizing the expected forward Kullback–Leibler divergence, they ensure an unbiased estimation of the score gradient, making SIVI more efficient in high-dimensional settings. Authors provide detailed derivations of their proposed methods with appropriate proofs and algorithms. Experimental results show that their approach outperforms or matches state-of-the-art SIVI methods, advancing both theoretical understanding and practical implementation of variational inference.

**Claims And Evidence:**

Good.

**Essential References Not Discussed:**

/

**Experimental Designs Or Analyses:**

* For experiment 1, there is no comparison with other methods. Maybe it is unnecessary since all methods might doing well on this.
* For experiment 2 and 3, although there are comparisons with other methods, I didn't find a clear benefit of the proposed methods. From the results and the figures, it seems like all methods are doing equally well. Then what are the real benefits of the proposed methods?

**Methods And Evaluation Criteria:**

Yes.

**Other Comments Or Suggestions:**

* A lot of $\in$ should be $\subseteq$ in this paper.
* Eq. (2) $f_\phi(\boldsymbol \epsilon) \sim q_{\boldsymbol y}$ is a confusing equation. Should be $y = f_\phi(\boldsymbol \epsilon)$.

**Other Strengths And Weaknesses:**

* The method derivation in this paper is good and solid.
* The visualization of the experiment part is clear.

**Questions For Authors:**

* For experiment 2, authors plot the measured correlation of $\beta$ in Fig. 3. What does a negative correlation mean? If the authors want to show that after matching with the true parameter of SGLD, the authors should plot the learned $\beta$ and the true $\beta$ after matching and showing that they form a diagonal line. But what authors plot is a pairwise correlation of the learned $\beta$ and the true $\beta$. If plotting correlation, I would hope to see more close to 1 correlations, after matching the learned with the true. Could authors explain more about this figure and what it implies. Thanks!
* In table 1, a close to 0 KL divergence means a good approximated distribution. However, the overall magnitude of the KL divergence may vary, depending on the nature of the true distribution. Is there a way to normalize the KL divergence? Otherwise, people don't know whether a number like 0.8 is close to 0 or not good enough, which will not be helpful compared with a direct visual comparison.

**Relation To Broader Scientific Literature:**

/

**Theoretical Claims:**

Glanced at the derivations but not into their details.

---

> ### Author Rebuttal · Authors · 2025-04-01
>
> 1. **Experiment 1 Comparison**: We agree that no comparison is needed for Experiment 1, as it primarily serves as a sanity check rather than a competitive benchmark.
>
> 2. **Correlation in Experiment 2**: We computed all possible correlations separately for the true $\beta$ and the estimated $\beta$, considering each coordinate individually. A negative correlation, $\rho_{ij}$, means that $\beta_i$ and $\beta_j$ (where $\beta$ represents here for example the true $\beta$ in this case) are negatively correlated. After obtaining these correlations for both the true and estimated $\beta$, we matched them so that the best possible outcome would form a diagonal line, ensuring alignment between the learned and true $\beta$. In Eq. 36, this can also be seen since the correlations are always computed with respect to a fixed parameter vector. We will adapt the manuscript to make this process clearer for the reader. Our results demonstrate the state-of-the-art performance among SIVI variants. We will clarify this point again in the paper and thank the reviewer for the comment.
>
>
> 3. **Experiment 3 Evaluation**:
> > “From the results and the figures, it seems like all methods are doing equally well.”
>
> In Experiment 3, we included a [table](https://figshare.com/s/f45697639e8908df84dd?file=53348411) demonstrating that only our proposed method (AISIVI) performs on par with the state-of-the-art KSIVI method. Additionally, we have improved the [figure](https://figshare.com/s/f45697639e8908df84dd?file=53348420) to better highlight that AISIVI is the only method capturing variance as effectively as KSIVI.
>
> 4. **Code Availability**: We have inquired with the area chair about the permissibility of providing the code currently hosted on (Anonymous) Github (but the rebuttal rules only allow figures and tables to be linked).
>
> 5. **Notation and Equation Clarifications**: The use of subset notation (⊆) is indeed correct, given our definition of $\mathcal{R}$. However, we recognize that this notation might lead to misunderstandings and will adapt it to a more standard form. We have also incorporated the suggested clarification for Eq. (2).
>
> 6. **KL Divergence Normalization**: As far as we know, there is no standard way to normalize KL divergence. However, to provide better intuition, we have added a [figure](https://figshare.com/s/f45697639e8908df84dd?file=53348417) depicting different training states alongside their KL values, helping to contextualize the reported numbers.
>
>
> We hope these clarifications address your concerns and demonstrate our commitment to improving the manuscript. Thank you again for your valuable feedback. Should the reviewer find the response satisfactory, we would appreciate reconsidering the initial score. Otherwise, we remain fully committed to addressing any remaining concerns during the second author response phase.

---

> > ### Comment · Reviewer_Ssns · 2025-04-05
> >
> > Thanks for the authors' rebuttal, I'm not very confident, but most of my concerns have been addressed, and I have raised my score from 2 to 3.

---

### Decision · Program_Chairs · 2025-05-01

**Decision:**

Accept (poster)

**Comment:**

The authors develop a new approach for semi-implicit variational inference, where the variational distribution marginalizes an implicit distribution with known likelihood. Optimizing the variational bound with pathwise estimators requires the gradient of log of q. The fresh direction is to explore how to estimate this gradient with importance sampling. A through analysis with consideration to memory usage is also carried out.